# Neural Diffusion Distance for Image Segmentation

**Jian Sun and Zongben Xu**
School of Mathematics and Statistics
Xi'an Jiaotong University, P. R. China
`{jiansun,zbxu}@xjtu.edu.cn`

## Abstract

Diffusion distance is a spectral method for measuring distance among nodes on graph considering global data structure. In this work, we propose a *spec-diff-net* for computing diffusion distance on graph based on approximate spectral decomposition. The network is a differentiable deep architecture consisting of feature extraction and diffusion distance modules for computing diffusion distance on image by end-to-end training. We design low resolution kernel matching loss and high resolution segment matching loss to enforce the network's output to be consistent with human-labeled image segments. To compute high-resolution diffusion distance or segmentation mask, we design an up-sampling strategy by feature-attentional interpolation which can be learned when training spec-diff-net. With the learned diffusion distance, we propose a hierarchical image segmentation method outperforming previous segmentation methods. Moreover, a weakly supervised semantic segmentation network is designed using diffusion distance and achieved promising results on PASCAL VOC 2012 segmentation dataset.

## 1 Introduction

Spectral analysis is a popular technique for diverse applications in computer vision and machine learning, such as semi-supervised learning on graph [39], image segmentation [17, 31], image matting [21], 3D shape analysis [36], etc. Spectral clustering and diffusion distance are two typical spectral techniques that rely on affinity matrix over a graph. By decomposing the affinity matrix using spectral decomposition, the corresponding eigenvectors encode the global structure of data, and can be utilized for spectral clustering, diffusion distance computation, image segmentation, etc.

Computing affinity matrix on graph for identifying the relations of each node w.r.t. other nodes is a fundamental task with potential applications in image segmentation [31], interactive image labeling [11] , object semantic segmentation [18, 22], video recognition [35], etc. Traditionally, the affinity matrix is either based on hand-crafted features [11, 31] or directly computed based on pairwise feature similarity of graph nodes without considering global structure of underlying graph [35, 37].

In this work, we propose *neural diffusion distance* (NDD) on image inspired by diffusion distance [7, 8], which is a spectral method for computing pairwise distance considering global data structure by spectral analysis. We propose to compute neural diffusion distance on image using a novel deep architecture, dubbed as *spec-diff-net*. This network consists of a feature extraction module, and a diffusion distance module including the computations of probabilistic transition matrix, spectral decomposition and diffusion distance, in an end-to-end trainable system.

To enable computation of spectral decomposition in an efficient and differentiable way, we use simultaneous iteration [12, 32] for approximating the eigen-decomposition of transition matrix. Since the neural diffusion distance is computed on the feature grid with lower resolution than full image, we propose a learnable up-sampling strategy in spec-diff-net using feature-attentional interpolation for interpolating diffusion distance or segmentation map. The spec-diff-net is trained to constrain that

its output neural diffusion distance should be consistent with human-labeled segmentation masks using Berkeley segmentation dataset (BSD) [28].

We apply neural diffusion distance to two segmentation tasks, i.e., hierarchical image segmentation and weakly supervised semantic segmentation. For the first task, we design a hierarchical clustering algorithm based on NDD, achieving significantly higher segmentation accuracy. For the second task, with the NDD as guidance, we propose an attention module using regional feature pooling for weakly supervised semantic segmentation. It achieves state-of-the-art semantic segmentation results on PASCAL VOC 2012 segmentation dataset [23] in weakly supervised setting.

Our contributions can be summarized as follows. First, a novel neural diffusion distance and its deep architecture were proposed. Second, with neural diffusion distance, we designed a novel hierarchical clustering method and a weakly supervised semantic segmentation method, achieving state-of-the-art performance for image segmentation. Moreover, though we learn NDD on image, it can also be potentially applied to general data graph beyond image, deserving investigation in the future.

## 2   Related works

Traditional spectral clustering [26] or diffusion distance [25] rely on hand-crafted features for constructing affinity matrix. In [11], diffusion distance was computed based on color and textures. It was taken as the spatial range for applying image editing. In [1], a learning-based method was proposed for spectral clustering by defining a novel cost function differentiable to the affinity matrix.

Recently, spectral analysis was combined with deep learning. Spectral network [3] is a pioneering network extending conventional CNN on grid to graph by defining convolution using spectral decomposition of graph Laplacian. The affinity matrix and its spectral decomposition are precomputed. Diffusion net [24] is defined as an auto-encoder for manifold learning. The encoding procedure maps high-dimensional dataset into a low dimensional embedding space approximating diffusion maps, and the decoder maps from embedding space back to data space. Similarly, [2, 30] learn a mapping from data to its eigen-space of graph Laplacian matrix, then cluster the data by spectral clustering. The affinity matrix is separately learned by a siamese network in [30]. These networks were applied to toy datasets for data clustering. The most similar work to ours is [17], in which an end-to-end learned spectral clustering algorithm was proposed based on subspace alignment cost which is differentiable to feature extractor using gradients of SVD / eigen-decomposition. This deep spectral network was successfully applied to natural image segmentation.

Another category of related research is deep embedding methods that directly measure the distance / similarity of pixels in the deep embedded feature space [4, 5, 6, 14, 19]. For example, [5, 6] learned the embedding feature space and relied on metric learning to learn similarity of paired pixels for video segmentation. Compared with them, our neural diffusion distance also works in embedded feature space, but measures pixel distance by diffusion on graph in a concept of diffusion distance, and distances are computed in the eigen-space of transition matrix (i.e., diffusion maps). This results in more smooth and continuous diffusion distance maps for image, as will be shown in experiments.

Our proposed neural diffusion distance bridges diffusion distance and deep learning in an effective way. Compared with traditional diffusion distance [7, 8, 25], NDD is based on an end-to-end trainable deep architecture with learned features and hyper-parameters. Compared with (deep) spectral clustering [17, 26], our segmentation method is built based on NDD considering global image structure when measuring affinity of image pixels. As shown in experiments, NDD enables state-of-the-art results for image segmentation and weakly supervised semantic segmentation.

## 3   Diffusion map and diffusion distance

We first briefly introduce the basic theory of diffusion distance [7, 8, 11] on a graph. Given a graph $G = (V, E)$ with $N$ nodes $V = \{v_1, v_2, \cdots, v_N\}$ and edge set $E$. Assume that $\mathbf{f}_i$ is the feature vector of node $i$ $(i = 1, 2, \cdots, N)$. We first define similarity matrix $W$ with each element $w_{ij}$ as

$$w_{ij} = \exp(-\mu||\mathbf{f}_i - \mathbf{f}_j||_2^2), \text{ for } j \in S_N(i), \tag{1}$$

where $S_N(i)$ is neighborhood set of $i$. Then the probabilistic transition matrix $P$ can be derived by normalizing each row of $W$:

$$P = D^{-1}W, \text{ where } D = \text{diag}(W\vec{1}). \tag{2}$$

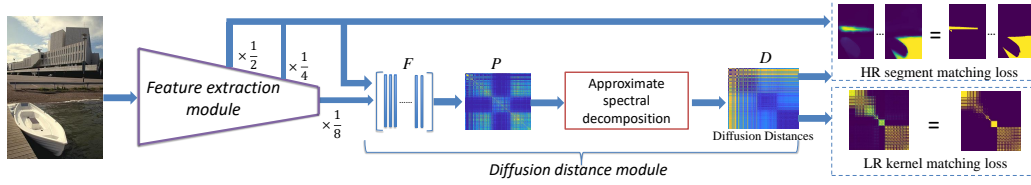

Figure 1: The spec-diff-net consists of a feature extraction module, followed by diffusion distance module, successively computing transition matrix, approximate spectral decomposition and diffusion distance. It is trained using HR segment matching loss and LR kernel matching loss.

Each element $P_{ij}$ of $P$ is the probability of a random walker walking from node $i$ to node $j$, and the $(i,j)$-th element of $P^t$ reflects the probability to move from a node $i$ to $j$ in $t$ time steps. Diffusion distance $D_t(i,j)$ is defined as sum of squared difference between the probabilities that random walker starting from two nodes $i, j$ and end up at a same node in the graph at time $t$:

$$D_t(i,j) = \sum_k (p(k,t|i) - p(k,t|j))^2 \tilde{w}(k), \tag{3}$$

where $p(k,t|i)$ is the probability that a random walk starting from node $i$ and end-up at node $k$ in $t$ time steps, and $\tilde{w}(k)$ is the reciprocal of the local density at node $k$. The diffusion distance will be small if there is a large number of short paths connecting these two points. Moreover, as $t$ increases, the diffusion distance between two nodes will decrease. The diffusion distance considers the global data structure and is more robust to noises compared with geodesic distance [7].

Suppose that $P$ has a set of $N$ eigenvalues $\{\lambda_m\}_{m=0}^{N-1}$ with decreasing order, and the corresponding eigenvectors are $\Phi_0, \cdots, \Phi_{N-1}$. When the graph has non-zero connections between each pair of nodes, the eigenvalues satisfy that $1 = \lambda_0 \geq \lambda_1 \geq \cdots \geq \lambda_{N-1}$. Then the diffusion distance is

$$D_t(i,j) = \sum_{m=0}^{N-1} \lambda_m^{2t} (\Phi_m(i) - \Phi_m(j))^2, \tag{4}$$

which is Euclidean distance in embedded space spanned by *diffusion maps*: $\lambda_0^t \Phi_0, \cdots, \lambda_{N-1}^t \Phi_{N-1}$.

## 4   Learning neural diffusion distance on image

We next design a deep architecture, dubbed as *spec-diff-net*, to compute diffusion distance by concatenating feature extraction and diffusion distance computation in a single pipeline.

### 4.1   Network architecture

As shown in Fig. 1, given an input image $I$, spec-diff-net successively processes the image by feature extraction module and diffusion distance module consisting of computations of transition matrix, eigen-decomposition and diffusion distance. Its output is called *neural diffusion distance*, which is sent to training loss for end-to-end training.

**Feature extraction module.** For extracting features from image $I$, it consists of repetitions of convolution, ReLU and max-pooling layers. We denote this module as $f(I; \Theta)$ with network parameters $\Theta$, then its output is features $F \in R^{w \times h \times d}$ and can be reshaped to $R^{N \times d}$ ($N = w \times h$).

**Diffusion distance module.** Based on features $F$, this module first computes transition matrix $P = D^{-1}W$, $W = \exp(-\mu||\mathbf{f}_i - \mathbf{f}_j||^2)$, and $\mathbf{f}_i$ is feature of $i$. Then it computes eigen-decomposition of $P$ as discussed in sect. 4.2. Suppose $\Lambda = \{\lambda_1, \cdots, \lambda_N\}$ and $\Phi$ are eigenvalues and matrix of eigenvectors, then the diffusion distance between $i$ and $j$ on feature grid can be computed by Eq. (4).

### 4.2   Approximation of spectral decomposition

An essential component in spec-diff-net is spectral decomposition of transition matrix $P \in R^{N \times N}$. The complexity of its spectral decomposition is commonly $O(N^3)$. For better adapting to larger $N$,

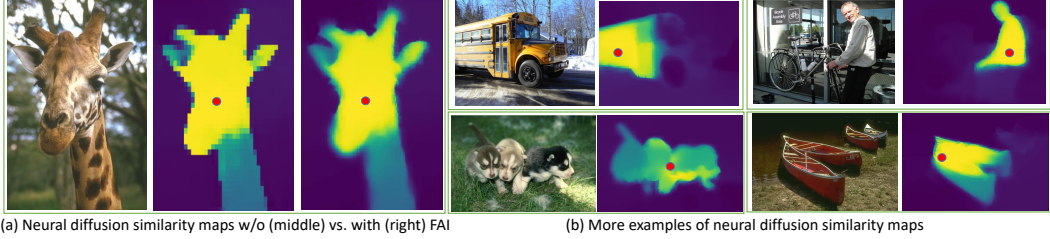

(a) Neural diffusion similarity maps w/o (middle) vs. with (right) FAI       (b) More examples of neural diffusion similarity maps

Figure 2: Neural diffusion similarity maps of image pixels indicated by red dots. In (a), the middle and right images are neural diffusion similarity w/o and with feature-attentional interpolation (FAI).

we design a differentiable approximation of spectral decomposition based on simultaneous iteration algorithm [12, 32], which is an extension of power iteration to approximately compute a set of $N_e$ dominant eigenvalues and eigenvectors of a matrix. The algorithm initializes $N_e$ dominant eigenvectors by a matrix $U_0$ in size of $N \times N_e$, then iteratively runs

$$Z_{n+1} = PU_n, \ \{U_{n+1}, R_{n+1}\} = \mathrm{QR}(Z_{n+1}), n = 0, \cdots, T, \tag{5}$$

where QR stands for QR-decomposition. It can be proved that, as $n \to \infty$, $U_n$ and diagonal values of $R_n$ respectively approximate the dominant $N_e$ eigenvectors and corresponding eigenvalues.

As shown in Eq. (4), we aim to compute eigenvectors together with powered version of eigenvalues $\lambda^{2t}$ of $P$. We therefore utilize simultaneous iteration algorithm to compute spectral decomposition of $P^{2t}$, i.e., taking $P^{2t}$ to substitute $P$ in Eq. (5). The following proposition shows that this simple revision (we call it accelerated simultaneous iteration) can improve the convergence rate.

***Proposition 1.*** Assume eigenvalues of $P$ satisfy $\lambda_0 > \lambda_1 > \cdots > \lambda_{N_e-1} > \lambda_{N_e}$, and all leading principal sub-matrices of $\Gamma^T U_0$ ($\Gamma$ is a matrix with columns $\Phi_1, \cdots, \Phi_{N_e}$) are non-singular, then columns of $U_n$ converge to top $N_e$ eigenvectors in linear rate of $(\max_{k \in [1,N_e]} \{|\lambda_k|/|\lambda_{k-1}|\})^{2t}$, and diagonal values of $R_n$ converge to corresponding top $N_e$ eigenvalues $\lambda_0^{2t}, \cdots, \lambda_{N_e-1}^{2t}$ in same rate.

Please see supplementary material for its proof. By approximating spectral decomposition of $P^{2t}$ instead of $P$, convergence rate is improved from linear rate of $\max_{k \in [1,N_e]} \{|\lambda_k|/|\lambda_{k-1}|\}$ to $\max_{k \in [1,N_e]} \{(|\lambda_k|/|\lambda_{k-1}|)^{2t}\}$ if $t > 0.5$. Since computational complexity of QR decomposition is $O(N_e N^2)$, then that of simultaneous iteration is $O(T N_e N^2)$. As discussed later, we only retain top $N_e \ll N$ ($N_e = 50$) eigenvalues, and truncate iterations $T$ ($T = 2$), therefore, the complexity $O(T N_e N^2)$ is smaller than original eigen-decomposition in $O(N^3)$ when $N$ is large.

### 4.3 Up-sampling by feature-attentional interpolation

The diffusion distance is computed on the feature grid of $F$ which is in lower-resolution compared with input image, we therefore design an interpolation method to up-sample the diffusion distance map (or segmentation map). The feature extractor in spec-diff-net can output multi-scale features $F^0, \cdots, F^L$ by its intermediate layers with feature grids of $\Omega^0, \cdots, \Omega^L$ from high resolution to low resolution. We interpolate a map $y^L$ from coarsest to finest level step by step. Suppose we already have the map $y^l$ at level $l$, we interpolate it to the finer level $l-1$ by feature-attentional interpolation:

$$y_i^{l-1} = \sum_{j \in \tilde{\Omega}^l \cap S_{at}(i)} \frac{1}{Z_i^{l-1}} \exp(-\gamma ||\mathbf{f}_i^{l-1} - \mathbf{f}_j^{l-1}||^2) y_j^l, \ \ i \in \Omega^{l-1}, \tag{6}$$

where $Z_i^{l-1} = \sum_{j \in \tilde{\Omega}^l \cap S_{at}(i)} \exp(-\gamma ||\mathbf{f}_i^{l-1} - \mathbf{f}_j^{l-1}||^2)$ is the normalization factor, $S_{at}(i)$ is a region neighboring pixel $i$, $\tilde{\Omega}^l$ is the grid by up-scaling grid coordinates of $\Omega^l$ to the finner scale coordinate system of $\Omega^{l-1}$, $j \in \tilde{\Omega}^l \cap S_{at}(i)$ is a point in $\tilde{\Omega}^l$ neighboring $i$ at $(l-1)$-th level, and $\mathbf{f}_j^{l-1}$ is its corresponding feature which is bi-linearly interpolated if it is not at integer coordinates. In this way, each pixel of up-sampled map $y^{l-1}$ is the weighted combination of values of its neighboring pixels up-sampled from lower-resolution grid, and the weights are computed based on feature similarity. All the computations are differentiable, and will be incorporated into spec-diff-net as discussed in sect. 5.

# 5 Network training for learning neural diffusion distance

We train spec-diff-net on image by enforcing its output, i.e., neural diffusion distance, to be consistent with human labeled segmentations in training set. Please see Fig. 2 for examples of learned neural diffusion distance (similarity). We define two training losses to learn neural diffusion distance.

**Low-resolution (LR) kernel matching loss.** Given output neural diffusion distance matrix $D_t$ with element measuring diffusion distance of paired pixels, we first transform it to *neural diffusion similarity* matrix $K_D = \exp(-\tau D_t)$. Then this loss enforces that $K_D$ measuring similarities of paired pixels at low resolution feature grid should be consistent with $K_{gt}$ defined by human-labeled segmentation, i.e., $(i, j)$-element of $K_{gt}$ is 1 if $i, j$ are in a same segment, and zero otherwise. Then we define the LR kernel matching loss as

$$L_{lr}(K_D, K_{gt}) = - \langle K_D/||K_D||_F, K_{gt}/||K_{gt}||_F \rangle. \tag{7}$$

**High-resolution (HR) segment matching loss.** We define *neural diffusion similarity map* of pixel $i$ as $i$-th row of $K_D$ (denoted as $K_D^i$) measuring similarities of $i$ with remaining pixels. We enforce that neural diffusion similarity map of each pixel $i$ is consistent with labeled segmentation mask at image resolution. To reduce training overhead, we randomly select pixel set $S$ including one sample for each segment in human labeled segmentation, then high-resolution segment matching loss is

$$L_{hr}(K_D, \hat{K}_{gt}) = \sum_{i \in S} - \left\langle \hat{K}_D^i/||\hat{K}_D^i||, \hat{K}_{gt}^i/||\hat{K}_{gt}^i|| \right\rangle, \tag{8}$$

where $\hat{K}_{gt}$ is the ground-truth human-labeled similarity matrix at image resolution, $\hat{K}_D^i = \text{UpSample}(K_D^i)$ and "UpSample" denotes the feature-attentional interpolation discussed in sect. 4.3. We use three-scales features with 1/2, 1/4, 1/8 factors of input image width and height for interpolation, and these features are outputs of conv1, conv2, conv5 of ResNet-101 [15]. $K_D, K_{gt}, \hat{K}_{gt}$ are all with elements in $[0, 1]$ and ones on their diagonals, therefore it is easy to verify that $L_{lr}$ and $L_{hr}$ are minimized when their two variables, i.e., similarity matrices, are exactly same.

**Training details.** The spec-diff-net is a deep architecture with differentiable building blocks. We train it on BSD500 dataset [28] by auto-differentiation, and each image has multiple human labeled boundaries. From these boundaries, each image can be segmented into regions. Compared with semantic segmentation labels, the segmentation labels of BSD500 do not indicate semantic categorization for pixels, and only indicate that pixels in a segment are grouped based on human's observation. To speed up the training process, we first pre-train our spec-diff-net using LR kernel matching loss, then add the HR segment matching loss which is more computational expensive due to the up-sampling by feature-attentional interpolation. We use ResNet-101 (excluding classification layer) pre-trained on MS-COCO [33] as in [20] for feature extraction and train spec-diff-net in 160000 steps. Since components of spec-diff-net are differentiable, we learn parameters $\Theta$ of features extractor, $\mu, t, \gamma$ in Eqs. (1,4,6), and $\tau$ in $K_D$. We empirically found that eigenvalues of transition matrix $P$ decrease fast from maximal value of one, we therefore set $N_e = 50$ in approximation of spectral decomposition for covering dominant spectrum. $U_0$ in simultaneous iteration is initialized by $N_e$ columns of one-hot vectors with ones uniformly located on feature grid. The neighborhood width when computing $W$ in Eq. (1) is set to 17 on feature grid. It takes 0.2 seconds to output neural diffusion distance for an image in size of $321 \times 481$ on a GeForce GTX TITAN X GPU.

**Illustration of diffusion distance.** Figure 2 illustrates examples of learned diffusion similarity maps with respect to the pixels on image indicated by red points. Figure 2(a) shows that feature-attentional interpolation can up-sample neural diffusion similarity maps without aliasing artifacts. We also tried a siamese network using Resnet-101 backbone as ours to learn pairwise similarity in embedded feature space (denoted as "Embedding"), and it can be seen that our neural diffusion distance is smooth and continuous, compared with "Embedding" method.

**Effects of parameters in approximate spectral decomposition.** Table 1 presents training (300 images in "train + val" of BSD500 dataset) and test (200 images in "test" of BSD500 dataset) accuracies measured by cosine similarity of estimated neural diffusion similarity matrix $K_D$ with target similarity matrix $K_{gt}$ using different hyper-parameter $T$ and initialized $t$ in approximate spectral decomposition. Note that simultaneous iteration serves as a differentiable computational

Table 1: Effects of different parameters in approximate spectral decomposition.

| $(T, t)$ | (1,5) | (1,10) | (2,5) | (2,10) | (3,5) | (3,10) |
|---|---|---|---|---|---|---|
| Train+val | 0.778 | 0.785 | 0.785 | **0.794** | 0.777 | 0.785 |
| Test | 0.701 | 0.709 | 0.738 | 0.741 | 0.735 | **0.748** |

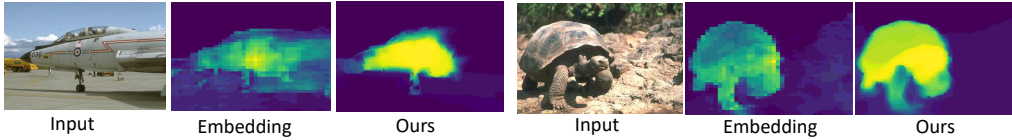

| Input | Embedding | Ours | Input | Embedding | Ours |

Figure 3: Visual comparison of similarity maps between deep embedding method and our neural diffusion distance. Each map shows the similarities w.r.t. the central pixel in the image.

block in spec-diff-net which is end-to-end trained for minimizing final training loss. We observe that, increasing initialization of $t$ from 5 to 10 and iterations $T$ from 1 to 2 all increase the training and test accuracies, but saturate after further increasing $T$ and initialized $t$. In the followings, we set $T = 2$ and initialize $t = 10$.

## 6 Application to hierarchical image segmentation

We first apply neural diffusion distance to image segmentation. We train spec-diff-net on BSD500 "train" and "val" sets, and test it on "test" set. Given a test image $I$, $K_D$ is its neural diffusion similarity matrix measuring neural diffusion similarity between pairs of grid points. With $K_D$, we design a hierarchical clustering algorithm for hierarchical image segmentation. The basic idea is to first identify a set of cluster centers, and then run the kernel $k$-means algorithm [9] with $K_D$ as the kernel to produce a finest segmentation of image. Then we gradually aggregate these segments to derive a hierarchy of image segmentations. To initialize the cluster centers, we iteratively add a new cluster center with its diffusion similarity map best covering the residual coverage map $1 - U_{cov}$ with $U_{cov} \in R^{N \times 1}$ initialized as zeros. Specifically, we iteratively add cluster center by:

$$i^* = \arg\max_i \{K_D^i (1 - U_{cov})\}, C = C \cup \{i^*\}, U_{cov} = \min\{U_{cov} + K_D^{i^*}, 1\}, \quad (9)$$

where $K_D^i$ is the $i$-th row of $K_D$, which is just the diffusion similarity map of $i$, and $C$ is the set of cluster centers. The iteration stops until the residual coverage map is smaller than a threshold

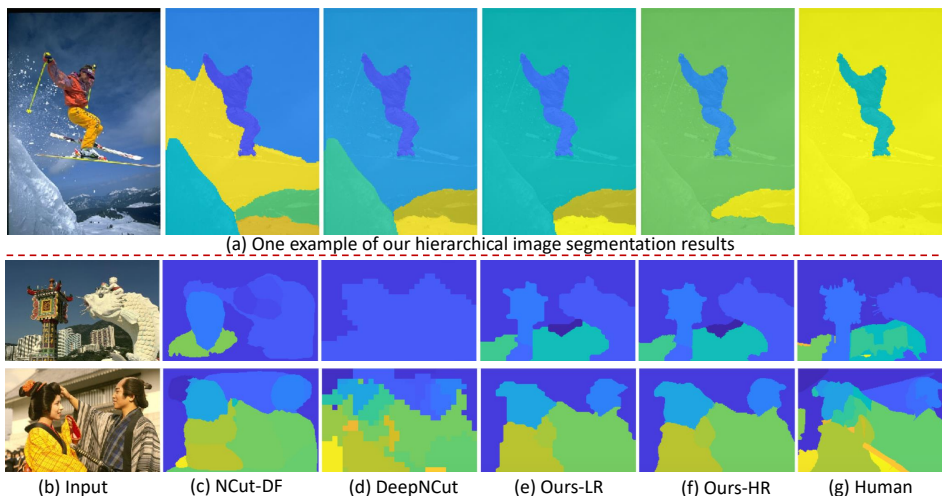

(a) One example of our hierarchical image segmentation results

| (b) Input | (c) NCut-DF | (d) DeepNCut | (e) Ours-LR | (f) Ours-HR | (g) Human |

Figure 4: Comparison of image segmentation results. (a) illustrates hierarchical image segmentation with decreasing number of segments. (b) compares segmentation results by different methods.

Table 2: Comparison of different segmentation methods.

| Methods | NCut [31] | NCut-DF | DeepNCut [17] | Ours-LR | Ours-HR |
|---------|-----------|---------|---------------|---------|---------|
| MAX | 0.53 | 0.56 | 0.70 | 0.78 | **0.80** |
| AVR | 0.44 | 0.48 | 0.60 | 0.68 | **0.69** |

(0.02) in average on pixels. After segmenting image $I$ to a set of segments with these initial centers by kernel $k$-means, we iteratively aggregate these segments by merging one pair of segments with largest average feature similarity in each step until achieving a single cluster for the whole image. In this way, we generate a hierarchy of segmentations with decreasing number of segments.

In Fig. 4, we illustrate an example of hierarchical image segmentation (Fig. 4(a)), and comparisons with other segmentation methods, including normalized cut [31] using deep feature (NCut-DF), deep normalized clustering (DeepNCut) [17], our methods w/o (Ours-LR) and with (Ours-HR) feature-attentional interpolation of segmentation masks. The quantitative comparisons are shown in Tab. 2. Accuracy is measured by average (AVG) and best (MAX) covering metric under optimal image scale criterion [28] as in [17]. Our algorithm achieves significantly better accuracies on "test" set of BSD500. For example, DeepNCut is a state-of-the-art deep spectral segmentation method based on differentiable eigen-decomposition, and our method achieves nearly 0.1 higher in accuracy.

# 7 Application to weakly supervised semantic segmentation

We also apply neural diffusion distance to weakly supervised semantic segmentation, i.e., learning to segment given an image set with only image-level classification labels. The basic idea is as follows. Since neural diffusion distance determines the similarities of each pixel w.r.t. other pixels on feature grid, which can be taken as spatial guidance for localizing where is the object of interest in a weakly supervised setting. Overall, we combine segmentation and classification in a single network, and train the network only using class labels. This is achieved by designing an attention module guided by diffusion distance to generate "pseudo" segmentation maps, which are utilized for computing global image features by weighted average pooling using weights based on "pseudo" segmentation. The global image features are taken as input of training loss to predict image class labels.

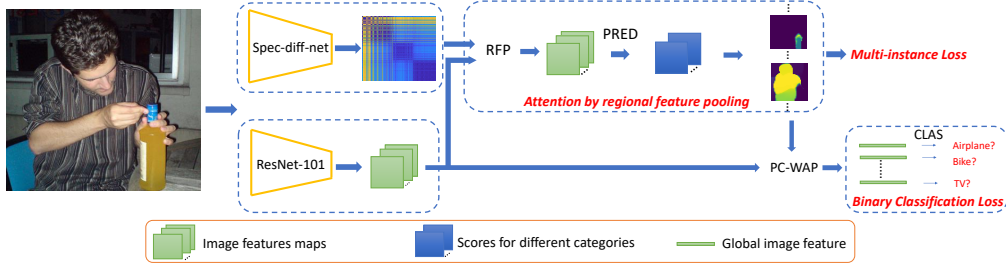

Figure 5: The architecture of our weakly supervised segmentation network.

As shown in Fig. 5, given image $I$, we compute neural diffusion distance and similarity matrix $K_D \in R^{N \times N}$ by spec-diff-net. We also use Resnet-101 to extract features $F \in R^{N \times d}$ from $I$. Then we design an attention module using regional feature pooling (RFP) to generate pseudo segmentation probability maps $P \in R^{N \times c}$ ($c$ is number of classes). With pseudo segmentation maps, we compute per-category global features $F^{gl}$ by per-category weighted average pooling (PC-WAP) of $F$. Then features of $F^{gl}$ are sent to training loss to predict image labels. We next introduce these components.

**Regional feature pooling (RFP)**. It performs average feature pooling over region determined by diffusion distance for each pixel. We first generate binary spatial regional mask for each pixel on feature grid, simply implemented in parallel for all pixels by thresholding diffusion similarity matrix $K_D$ by $M = \delta[K_D > \mu] \in R^{N \times N}$ ($\mu$ is initialized as 0.5, $\delta[\cdot]$ is binary with value of 1 if its variable is true). Then we average-pool features in regional mask of each pixel, which can be implemented by $F_M = \text{diag}((M\vec{1})^{-1})MF$, $F_M \in R^{N \times d}$. Therefore, for each pixel, this operation pools the features for each pixel over the region of pixels around it with neural diffusion similarities larger than $\mu$.

**Pseudo segmentation prediction (PRED)**. With the pooled features by RFP, we predict the per-pixel segmentation probabilities by classifier $\{H \in R^{d \times c}, b \in R^{c \times 1}\}$, i.e.,

$$P^{sg} = \text{Softmax}_{\text{cl}}(F_M H + \vec{1}b^T), \tag{10}$$

where $P^{sg} \in R^{N \times c}$, $\text{Softmax}_{\text{cl}}(\cdot)$ is softmax across different categories. Therefore, the $i$-th column of $P^{sg}$ indicates the probability map of pixels belonging to $i$-th category.

**Per-category weighted average pooling (PC-WAP).** Based on the "pseudo" segmentation probability maps in $P^{sg}$, we compute global image feature for $i$-th category by weighted average pooling:

$$F_i^{gl} = F^T[\text{Softmax}_{\text{sp}}(P_i^{sg}; \theta_i)], \quad \text{for } i = 1, \cdots, c, \tag{11}$$

where $P_i^{sg} \in R^{N \times 1}$ is the $i$-th column of $P^{sg}$, $\text{Softmax}_{\text{sp}}(P_i^{sg}, \theta_i) \in R^{N \times 1}$ is softmax operator conducted spatially over feature grid with temperature $\theta_i$. Different from global average pooling (GAP) in [38], we compute global image feature by weighted average pooling with weights based on "pseudo" segmentation probability maps in $P^{sg}$, indicating which pixels are relevant to each class.

**Training loss.** In weakly supervised setting, we only have image-level class labels, we therefore design training loss only with the guidance of class labels. Given the globally pooled features using PC-WAP, we predict the probabilities of image belonging to different categories (i.e., "CLAS" block in Fig. 5) by $P^{cl} = \{H_i^T F_i^{gl} + b_i\}_{i=1}^c$, where $H_i$ and $b_i$ ($i = 1, \cdots, c$) are respectively one column and element of $H, b$ in "PRED" block. Then training loss is defined by binary cross-entropy (BCE):

$$L_{ws} = \text{BCE}(P^{cl}, y^{cl}) + \text{BCE}(P_{max}^{sg}, y^{cl}), \tag{12}$$

where $P_{max}^{sg} \in R^c$ is a vector with elements as maximal values of columns of $P^{sg}$ over feature grid for different categories, therefore the second term is multiple instance loss. Minimizing $L_{ws}$ forces the classifier of $H, b$ to predict correct image-level labels and pixel-level segmentation implicitly.

Table 3: Comparison of different weakly supervised semantic segmentation methods.

| Methods | MIL [29] | Saliency [27] | RegGrow [16] | RandWalk [34] | AISI [10] | Ours |
|---------|----------|---------------|--------------|---------------|-----------|------|
| Val | 42.0 | 55.7 | 59.0 | 59.5 | 63.6 | **65.8** |
| Test | - | 56.7 | - | - | 64.5 | **66.3** |

Table 4: Comparison with baseline semantic segmentation methods.

| Methods | GAP [38] | Embedding | Ours (w/o RFP) | Ours (w/o sharing) | Ours |
|---------|----------|-----------|----------------|--------------------|------|
| Val | 45.2 | 54.7 | 44.6 | 64.7 | **65.8** |

We train weakly supervised segmentation network (spec-diff-net is fixed and pre-trained on 500 images of BSD500) on VOC 2012 segmentation training set with augmented data [13] using only image labels. After training the network, we derive pseudo segmentation maps for training images, which are taken as segmentation labels for training another ResNet-101 for learning to segment. We train the nets on $321 \times 321$ patches with fixed batch normalization as pre-trained ResNet-101 due to limited batch size. We apply trained segmentation net on "val" and "test" of VOC 2012 segmentation dataset. The network is applied to a test image in multiple scales (scaling factors of 0.7, 0.85, 1) with cropped overlapping $321 \times 321$ patches, and these segmentation probabilities are averaged as the final prediction.

Table 3 compares segmentation accuracies in mIoU with other weakly supervised segmentation methods: multiple instance learning (MIL) [29], saliency-based method (Saliency) [27], region growing method (RegGrow) [16], random walking method (RandWalk) [34], and salient instances-based method (AISI) [10]. Note that RandWalk method [34] is based on random walking for label prorogation given human labeled scribbles. AISI [10] depends on the instance-level salient object detector trained on MS COCO dataset. We achieve 65.8% and 66.3% on "val" and "test" sets, which are higher than state-of-the-art AISI method also using ResNet-101 and same training set. Figure 6 shows examples of segmentation results (more results are in supplementary material).

*Ablation study:* As shown in Tab. 4, without regional feature pooling, i.e., ours (w/o RFP), the accuracy on "val" set decreases from 65.8 to 44.6. This shows that RFP is essential because it

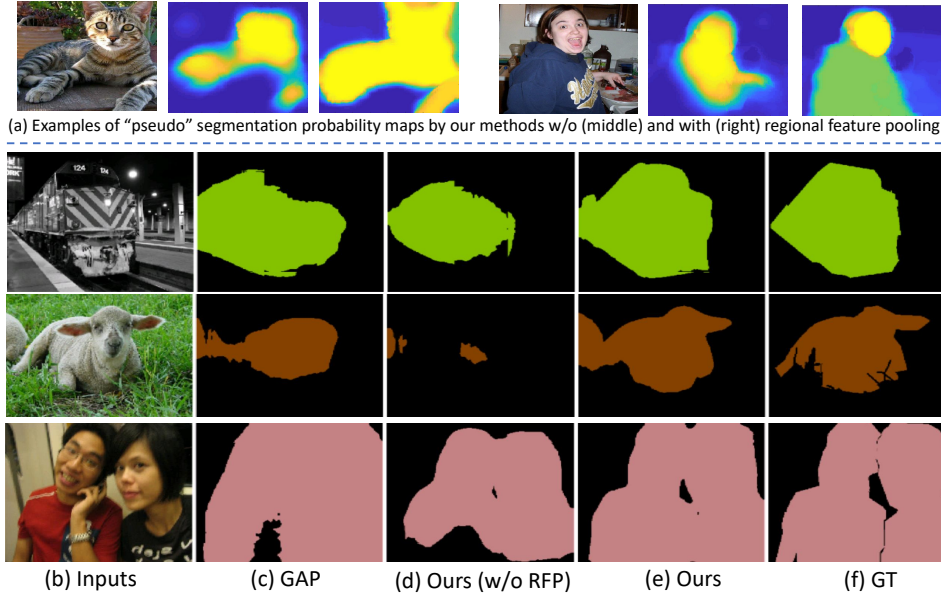

(a) Examples of "pseudo" segmentation probability maps by our methods w/o (middle) and with (right) regional feature pooling

(b) Inputs      (c) GAP      (d) Ours (w/o RFP)      (e) Ours      (f) GT

Figure 6: Examples of semantic segmentation results by different methods.

enforces that pixels with high neural diffusion similarities will have similar features, then they should be grouped and have similar segmentation probabilities. Furthermore, without sharing the classifiers for classification in training loss and segmentation in "PRED" module marginally decreases the result. When sharing classifiers, by optimizing the training loss, it jointly enforces that the classifier can predict global image class label and locations of objects of interest using the same classifier. In Tab. 4, we also report result using same weakly supervised segmentation architecture as ours but with similarity learned by embedding method, and the accuracy is significantly lower that our method based on diffusion distance.

## 8 Conclusion and future work

In this work, we proposed a novel deep architecture for computing neural diffusion distance on image based on approximate spectral decomposition and feature-attentional interpolation. It achieved promising results for hierarchical image segmentation and weakly supervised semantic segmentation. We are interested to further improve the neural diffusion distance, e.g., better handling transparent object boundaries, and apply it to more applications, e.g., image colorization, editing, labeling, etc.

**Acknowledgement.** This work was supported by National Natural Science Foundation of China under Grants 11971373, 11622106, 11690011, 61721002, U1811461.

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
