[Supplementary Material · diffMap_supp.pdf]

# Neural Diffusion Distance for Image Segmentation (Supplementary Material)

**Jian Sun and Zongben Xu**
School of Mathematics and Statistics
Xi'an Jiaotong University, P. R. China
{jiansun,zbxu}@xjtu.edu.cn

This supplementary material is organized as follows. We first give a brief proof of proposition 1. Then we will present more examples on the learned diffusion distance, image segmentation results, and weakly supervised semantic segmentation results.

## 1   Proof of proposition 1

For the transition matrix $P$, assume its spectral decomposition is $P = \Phi\Lambda\Phi^T$, then the powered version of $P$: $P^{2t} = \Phi\Lambda^{2t}\Phi^T$. It is known that, for a matrix $P$, the simultaneous iteration has the following conclusion [9, 27].

*By applying simultaneous iteration (Eq.(5)) to matrix $P$, assume eigenvalues of $P$ satisfy $\lambda_0 > \lambda_1 > \cdots > \lambda_{N_e-1} > \lambda_{N_e}$, and all leading principal sub-matrices of $\Gamma^T U_0$ ($\Gamma$ is a matrix with columns as $\Phi_1, \cdots, \Phi_{N_e}$) are non-singular, then columns of $U_n$ converge to leading $N_e$ eigenvectors in linear rate of $(\max_{k \in [1, N_e]}\{|\lambda_k|/|\lambda_{k-1}|\})$, and diagonal values of $R_n$ converge to corresponding top $N_e$ eigenvalues $\lambda_0, \cdots, \lambda_{N_e-1}$ in the same rate.*

$P^{2t}$ has the same eigenvectors as $P$, but has eigenvalues $\lambda_0^{2t}, \cdots \lambda_N^{2t}$ as powered version of eigenvalues of $P$. Therefore, by applying simultaneous iteration to $P^{2t}$ instead of $P$, it can be concluded that the accelerated simultaneous iteration satisfies the proposition 1 in the submitted paper.

## 2   More examples and results

**Figure 1**: examples of learned diffusion similarity maps for images in BSD500 test set using trained spec-diff-net on BSD500 train+val sets.

**Figure 2**: examples of hierarchical segmentation results for images in BSD500 test set using proposed hierarchical segmentation algorithm and trained spec-diff-net on BSD500 train+val sets.

**Figure 3**: examples for comparing different segmentation methods. Each method can generate a hierarchy of segmentations with different number of segments. Here we only show the segmentation in the hierarchy that best matches the ground-truth segmentation labeled by human.

**Figure 4**: examples to show the pseudo-segmentation probability maps generated by our weakly-supervised semantic segmentation net on VOC 2012 augmented training dataset.

**Figure 5**: examples for comparing weakly-supervised semantic segmentation results on val set of VOC 2012 segmentation dataset.

Figure 1: Examples of learned diffusion similarity maps for images in BSD500 test set using trained spec-diff-net on BSD500 train+val sets.

Figure 2: Examples of hierarchical segmentation results for images in BSD500 test set using proposed hierarchical segmentation algorithm and trained spec-diff-net on BSD500 train+val sets. On each row, the second to final sub-images are hierarchical segmentations with increasing number of segments.

(a) Input    (b) NCut-DF    (c) DeepNCut    (d) Ours-LR    (e) Ours-HR    (f) Human

Figure 3: Examples for comparing different segmentation methods (NCut-DF [26], DeepNCut [13], our method w/o (ours-LR) and with (ours-HR) feature-attentional interpolation). Each method can generate a hierarchy of segmentation with different number of segments. Here we only show the segmentation in the hierarchy best matching the ground-truth segmentation labeled by human.

Figure 4: Examples to show the pseudo-segmentation probability maps generated by our weakly-supervised semantic segmentation net on VOC 2012 augmented training dataset.

(a) Input images    (b) GAP    (c) Ours (w/o RFP)    (e) Ours (full)    (f) Ground-truth

Figure 5: Examples for comparing weakly-supervised semantic segmentation results on val of VOC 2012 segmentation dataset. (RFP: Regional feature pooling).