[Reviews · NeurIPS 2019]

Reviewer 1



The paper is clearly written, everything proposed in the paper makes sense and seems like a natural thing to do (I had been working on the same problem, so I am entirely in favor of the pursued direction). Still, the paper is missing comparisons and references to other works in the direction of representing image or video segmentation in terms of a continuous embedding, e.g.: Segmentation-aware convolutional networks using local attention masks, Adam W Harley, et al, ICCV 2017 Dense and low-rank gaussian crfs using deep embeddings, S Chandra, et al, ICCV 2017 S. Kong, et al, "Recurrent Pixel Embedding for Instance Grouping", CVPR 2018 Video object segmentation by learning location-sensitive embeddings, H Ci, et al, ECCV, 2018 X. Wang et al, Non-local Neural Networks, CVPR 2018 Y. Chen, J. Pont-Tuset, A. Montes, and L. Van Gool Blazingly Fast Video Object Segmentation with Pixel-Wise Metric Learning Computer Vision and Pattern Recognition (CVPR), 2018. This is not only important for giving credit to earlier works. A more crucial question in connection with these works, is whether the structured layer adds something on top of the ability of a cnn to compute embeddings for image segmentation. In experiments that I have been working on it has been really hard to beat a well-tuned, plain convnet trained with a siamese loss, and introducing a spectral normalization layer only added complications. It would be really useful if the authors could do this comparison on top of a strong baseline (e.g. the methods mentioned above) and indicate whether the resulting embeddings (=eigenvectors) are any better than those delivered from the original baselines. (the method of [13] is outdated by 4 years).

Reviewer 2



- The paper is well written and easy to follow. - The proposed idea (combining deep learning and diffusion distance) is clean and well motivated. - The proposed approach can (potentially, though no evidence about it) be applied in other graph applications beyond images. - My big concern is with respect to the evaluation the experimental results. From my understanding (L172), the features are initialized with a network pre-trained on MS-COCO. This means that the ResNet-101 features already embeds strong pixelwise semantic information (acquired with fully supervised training). Therefore it seems not a fair comparison with other method. - I also believe that much of implementation/training details are missing. Eg, what is the optimization algorithm used? What hyper-parameters?

Reviewer 3



I am not expert to evalute its originality and novelty to semantic segmentation as I am familiar with spectural analysis and diffusion. But my main concerns lie in the experiments. From the method part, I expect it is a general segmentation approach which should be evaluated on standard semantic segmentation dataset. But the experiments lack of such analysis severely. Even on weakly supervised segmentation dataset, it only compared with one baseline.

[Author Response · NeurIPS 2019]

Table 1: Extended "*Table 2*" in submitted paper. Segmentation results on BSD500 dataset.

| Methods | NCut | NCut-DF | DeepNCut | Ours-LR | Ours-HR | Embedding |
|---------|------|---------|----------|---------|---------|-----------|
| MAX | 0.53 | 0.56 | 0.70 | 0.78 | **0.80** | 0.77 |
| AVR | 0.44 | 0.48 | 0.60 | 0.68 | **0.69** | 0.67 |

Table 2: Extended "*Table 3*" in submitted paper. Results for weakly supervised semantic segmentation on VOC2012.

| Methods | MIL | Saliency | RegGrow | RandWalk | AISI | Ours | Embedding |
|---------|-----|----------|---------|----------|------|------|-----------|
| Val | 42.0 | 55.7 | 59.0 | 59.5 | 63.6 | **65.8** | 54.7 |
| Test | - | 56.7 | - | - | 64.5 | **66.3** | 54.9 |

We thank reviewers for their comments, and will carefully revise paper considering these comments.

*Q1 (R1): References and comparison with a baseline that learns embeddings only through a standard convnet.*

Thanks for recommending these papers on learning embeddings for pairwise distance computation and we will include
them in our paper. Different from directly computing pairwise distance of pixels, our diffusion distance measures
pixel distance by diffusion on graph in a concept of random walks, and distances are computed in the eigen-space of
transition matrix (i.e., diffusion maps). Figure 1 in this rebuttal compares examples of learned similarity by siamese
network using Resnet-101 backbone same as ours (denoted as "Embedding" in the following), and it can be seen that
our neural diffusion distance is smooth and continuous. In Tab.1 and Tab. 2 in this rebuttal (i.e., the extensions of Tab. 2
and Tab. 3 in submitted paper), we added a column in each table showing results with the "Embedding" substituting our
neural diffusion distance, it is shown that our neural diffusion distance produced better performance for hierarchical
segmentation (Tab. 1) and weakly supervised segmentation (Tab. 2).

Figure 1: Visual comparison of similarity maps between embedding method and our neural diffusion distance. Each map shows the similarities w.r.t. the central pixel in the image.

*Q2 (R2): Unfairness in comparison.* In Tab.2 of this rebuttal, the state-of-the-art method of AISI [7] also depends on
external MS-COCO segmentation labels in their approach. RandWalk [29] uses human-labeled scribbles when training
segmentation network. Saliency [22] method depends on saliency model trained with bounding box annotations of
MSRA dataset. We will give more details of these compared methods in paper for clarity.

*Q3 (R2): Implementation/training details.*

In all the training tasks in the paper, we use Adam optimization algorithm. The learning rates are set to 1e-6 for
training spec-diff-net, 3e-7 for weakly supervised network in Fig. 4, and 1e-6 for image segmentation network. It takes
160000, 20000, 20000 network updating steps for training the above three cases respectively. We will include more
implementation details in the paper.

*Q4 (R3): Originality / novelty.* Diffusion distance is a mathematically sound distance on graph as defined in sect. 3.
This paper is an novel try to take advantage of deep network to learn diffusion distance for segmentation by extending it
to "neural diffusion distance" using end-to-end training. Several technical novelties have been proposed to achieve
that goal, including the proposed spec-diff-net, approximate eigen-decompostion with its convergence analysis, two
training losses, attention-based upsampling. Moreover, we apply it to two segmentation tasks with novel designs of
corresponding kernel k-means based hierarchical segmentation algorithm and attention-based semantic segmentation
network. These novel techniques enable promising results for image hierarchical clustering and weakly supervised
semantic segmentation.

*Q5 (R3): Results on segmentation benchmark, compare with one baseline on weakly supervised segmentation.*

We respectfully disagree because we evaluate neural diffusion distance for hierarchical image segmentation on bench-
mark BSD500 dataset, and weakly supervised semantic segmentation on benchmark VOC2012 dataset. These two
datasets are popular for image segmentation. We extensively compared with different methods as shown in Tables 2, 3,
and different baselines by ablation study in Tables 1, 4 of the submitted paper.

[Meta-Review · NeurIPS 2019]

The authors propose a combination of diffusion distances with deep networks trained for image segmentation. Two of the reviewers felt that there were interesting methodological contributions meriting acceptance, while all three reviewers indicated that baselines in the experiments could be improved.